biomedical engineering/biomedical engineering

fluorescence endoscope, tumour, fast digital lock-in algorithm, image SNR

**Authors for correspondence:**
Shengzhao Zhang
e-mail: zsz1990@ahmu.edu.cn
Zhe Zhao
e-mail: zhaozhe@tjpu.edu.cn

# Enhancement of signal-to-noise ratio for fluorescence endoscope image based on fast digital lock-in algorithm

Huiquan Wang[1], Meng Hu[2], Fang Xia[1], Meng Guo[1], Shengzhao Zhang[3], Zhe Zhao[2], Guang Han[1] and Jinhai Wang[1]

[1]School of Life Sciences, and [2]School of Electronic and Information Engineering, Tiangong University, Tianjin 300387, People's Republic of China
[3]School of Biomedical Engineering, Anhui Medical University, Hefei 230032, People's Republic of China

HW, 0000-0002-7896-303X; MH, 0000-0001-5571-4332

In this paper, the signal-to-noise ratios (SNR) of two image channels were enhanced with the fast digital lock-in algorithm. In order to simultaneously improve the quality of white and fluorescence images obtained by fluorescence endoscope, and improve the SNR to achieve a better image processing effect, two sources of white light and near-infrared light of a fluorescence endoscope were modulated, then the acquired images were demodulated into white and fluorescence images. A fluorescent endoscope experimental platform was setup to acquire endoscopic images of a target dyed by indocyanine green. The experimental results showed that the SNR of white and fluorescent images without the lock-in algorithm were 36.56 dB and 33.47 dB, respectively. However, with the lock-in algorithm, the SNR of white and fluorescent images were 39.54 dB and 35.70 dB, respectively. The SNR of white and fluorescent images was increased by 8.2% and 6.7%, respectively, by appling the digital lock-in algorithm. Therefore, this novel fluorescence endoscope based on the fast digital lock-in algorithm can rapidly and simultaneously obtain two-channel images of white light and fluorescence, effectively enhance the SNR of white and fluorescent images, and improve the imaging quality.

## 1. Introduction

Compared with traditional surgery, minimally invasive surgery has the advantages of less bleeding, faster recovery, less

postoperative pain and scars [1]. Currently, the recognition and resection of tumours mainly depend on a visual inspection and digital examination by surgeons under white light during minimally invasive surgery treatment. However, it is difficult to accurately recognize and completely remove the tumour during surgery when the contrast of the tumour edge is low or the tumour is small [2]. Owing to the different absorption of fluorescent agents in tumour and healthy tissues, indocyanine green (ICG) has been Food and Drug Administration approved, which can guarantee the safety during the operation, near-infrared fluorescence-guided surgery can improve the success rate of early stage cancer detection, reduce the rate of missed detection, improve the accuracy of tumour resection and reduce the risk of cancer recurrence. Owing to the deep penetration depth of near-infrared light in biological tissues and the high resolution of near-infrared fluorescence imaging [3–9], near-infrared light can provide doctors with an accurate, objective and effective organizational structure and functional information in real-time during surgery. At present, fluorescence imaging technology is widely used in the endoscope, mainly by the time-sharing method to collect the white images and fluorescence images. The dual-channel fluorescence endoscopy imaging system proposed by Oh *et al.* [10] is illuminated by white light and near-infrared light, and the white images and fluorescence images are collected by a time-sharing method. In this way, on the one hand, the image acquisition frame rate is reduced, and cannot realize simultaneous imaging of white and fluorescence channels. On the other hand, the fluorescence signal is easily disturbed by noise, and the signal-to-noise ratio (SNR) is relatively low. A team of professor Gang Li from Tianjin University [11,12] proposed a fast digital lock-in algorithm based on the classic digital lock-in algorithm. This algorithm no longer needs multiplication in the positive cross-correlation operation, and the amplitude and phase of the signal can be calculated by the addition and subtraction of discrete time series. Using this fast digital lock-in algorithm greatly reduces the amount of calculation and data storage, and reduces the algorithm's requirements on the microprocessor. At the same time, the microprocessor does not need to generate a reference signal with the same frequency as the input signal, which reduces the burden on the microprocessor and improves the speed of algorithm realization. The digital lock-in algorithm has been widely used in the fields of magnetic [13,14], light [15,16] and electricity [17] with the advantages of improving the SNR and the system stability. Therefore, this study proposes a fluorescence endoscope imaging system based on the fast digital lock-in algorithm to improve the image SNR of the fluorescence endoscope without reducing the image acquisition frame rate.

In this paper, the MATLAB software is used to simulate the digital lock-in algorithm, in other words, modulation of the light sources and image demodulation process of the fluorescence endoscope system are simulated, and then the image SNR theoretical values with and without digital lock-in algorithm processing are calculated. Based on the simulation results, a fluorescence endoscope imaging system is constructed. The system uses white light as the bright field illumination source and 760 nm narrow-band near-infrared light as fluorescent excitation light, and both sources are modulated simultaneously. A camera is used to capture the images. Then white and fluorescent images are obtained using the fast digital lock-in algorithm, followed by image SNR analysis. The experimental results show that the fast digital lock-in algorithm can improve the image SNR, facilitate the subsequent image processing process and improve the image processing effect while maintaining the image acquisition frame rate.

# 2. Fast digital lock-in algorithm for fluorescence endoscopic images acquisition

## 2.1. Dual-channel digital lock-in algorithm

The fluorescence endoscope consists of a white image signal channel and a fluorescent image signal channel, where the fluorescence signal is weak. Therefore, in order to detect the weak fluorescence signal under strong background, this study uses the fast digital lock-in algorithm to modulate the white light and near-infrared light sources to improve the SNR of the image signal. The digital lock-in algorithm detects the target signals using the principle of cross-correlation detection. The principle is to perform spectrum shifting on the original signal and use a low-pass filter to filter out the noise to obtain a high SNR demodulated signal. The schematic diagram of the dual-channel digital lock-in algorithm is shown in figure 1.

It can be seen from figure 1 that the two-channel digital lock-in algorithm requires two carriers with different frequencies, that is, the modulation frequencies of the light sources are different. The phase-sensitive detector is a multiplier that can multiply the high correlation input signal and the reference

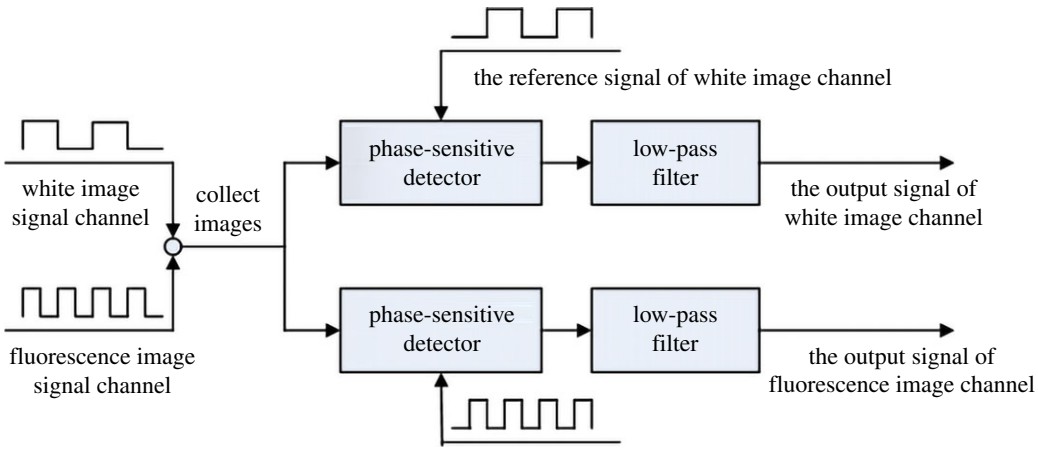

**Figure 1.** Schematic diagram of dual-channel digital lock-in algorithm.

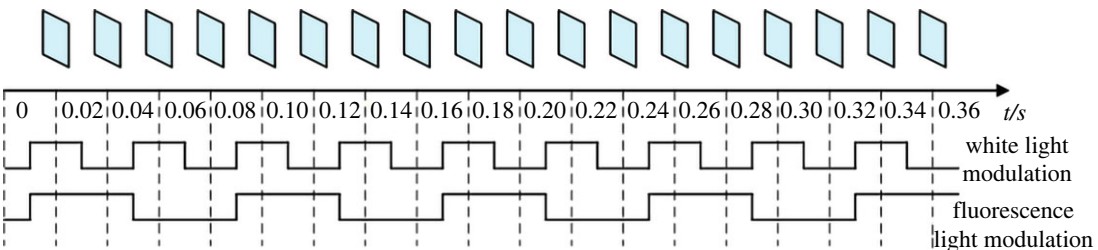

**Figure 2.** Schematic diagram of light sources modulation.

signal, and the original spectrum $\omega = \omega_0$ is moved to $\omega = 0$. Because the noise is not correlated with the reference signal, the noise spectrum will not migrate. After the input signal is multiplied with the reference signal, the low-pass filter is used to filter out the high-frequency noise, and the SNR of the image signal is improved. The modulation and the reference signals adopted in this paper are both square wave signals. The schematic diagram of light sources modulation is shown in figure 2.

Taking the white image signal channel as an example: assuming the original signal is $W$, the amplitude of the modulated square wave is $A$, the frequency is $\omega_0$, the Fourier series expansion is performed that can be expressed as

$$w(t) = \frac{4W}{\pi} \sum_{n=1}^{\infty} \frac{(-1)^{n+1}}{2n-1} \cos[(2n-1)\omega_0 t + \varphi]. \tag{2.1}$$

The Fourier series expansion of the reference signal is

$$s_w(t) = \frac{4A}{\pi} \sum_{n=1}^{\infty} \frac{(-1)^{n+1}}{2n-1} \cos[(2n-1)\omega_0 t]. \tag{2.2}$$

The low-pass filter can be regarded as an integrator, so the output signal through the low-pass filter can be expressed as

$$u_w(t) = \frac{1}{T} \int_0^{T_0} w(t) s_w(t) \, \mathrm{d}t. \tag{2.3}$$

In equation (2.3), if the integration time is long enough, the result of the integration is the average of the modulated and the reference signals. As the phase difference changes, the relationship between $u_0(t)$ and $\varphi$ can be expressed as

$$u_0(\varphi) = \left\{ \begin{array}{l} WA\left(1 - \dfrac{2\varphi}{\pi}\right), 0 < \varphi < \pi \\[2mm] WA\left(\dfrac{2\varphi}{\pi} - 3\right), \pi < \varphi < 2\pi \end{array} \right\}. \tag{2.4}$$

It is known from equation (2.4) that when the modulated and the reference signals are both square waves, the output signal is proportional to the original signal, and the white image signal and the fluorescent image signal can be detected.

## 2.2. Image signal-to-noise ratio

The image SNR is an important standard for the objective evaluation of image quality of fluorescence endoscopic imaging systems. The image SNR is the ratio of the power spectrum of the signal and the noise. The errors between the original and the processed images with and without the digital lock-in algorithm are calculated to determine the SNR enhancement effect of the digital lock-in algorithm.

The calculation formula of the image SNR is as follows:

$$I_{\text{original}} = \frac{1}{MN} \sum_{i=1}^{M} \sum_{j=1}^{N} f^2(i,j), \tag{2.5}$$

$$\text{MSE}_d = \frac{1}{MN} \sum_{i=1}^{M} \sum_{j=1}^{N} (f(i,j) - g_1(i,j))^2, \tag{2.6}$$

$$\text{MSE}_0 = \frac{1}{MN} \sum_{i=1}^{M} \sum_{j=1}^{N} (f(i,j) - g_0(i,j))^2, \tag{2.7}$$

$$\text{SNR}_d = 10 \log_{10} \frac{I_{\text{original}}}{MSE_d} \tag{2.8}$$

and

$$\text{SNR}_0 = 10 \log_{10} \frac{I_{\text{original}}}{MSE_0}. \tag{2.9}$$

In the equations (2.5), (2.6) and (2.7), $M$ and $N$ represent the numbers of rows and columns of the image, that is, the image size, respectively, $(i, j)$ represent the pixels in the image, $f$, $g_1$, and $g_0$ represent the original image, the image with the lock-in algorithm and the image without the lock-in algorithm, respectively. $\text{SNR}_d$ and $\text{SNR}_0$ represent the SNR of the images obtained with and without the lock-in algorithm, respectively.

# 3. Fluorescence endoscope image acquisition system

The schematic diagram of the fluorescence endoscope imaging system is shown in figure 3. The modulation signal generator is used to modulate the white light and the near-infrared light sources. The object to be measured is irradiated with white light and near-infrared light sources. The white image and the fluorescence image signals enter the fluorescence endoscopy guiding system and the image signals are collected by a complementary metal oxide semiconductor (CMOS). Then, the white and the fluorescent images are obtained by the image demodulator and displayed on a PC. Based on the system structure, a fluorescence endoscope imaging system was built, as shown in figure 4.

# 4. Results and discussion

## 4.1. Simulation experiment results and analysis

In order to objectively evaluate the effect of the digital lock-in algorithm on the image SNR of the fluorescence endoscopy imaging system, the white original image and the fluorescence original image were collected separately in this study, and the fluorescent agent ICG was applied to the two letters 'TJ' in 'TJPU' in the original image. MATLAB was used to conduct the simulation research of the digital lock-in algorithm. The original white original image and the fluorescent original image were copied to generate multiple frames of images, and the noise was added to the image to simulate the noise in the actual image signal. Figure 5 shows the original images, the images with the digital lock-in algorithm and images without the digital lock-in algorithm when SNR = 10.46 dB.

Figure 5a and b represent the original white and fluorescent images, respectively. Figure 5c and d represent the white and the fluorescence images after the digital lock-in algorithm processing, respectively, while, figure 5e and f represent the white and the fluorescent images that have not been processed by the digital lock-in algorithm, respectively. It can be seen from figure 5 that the image quality after lock-in algorithm processing is significantly improved, compared with the image quality

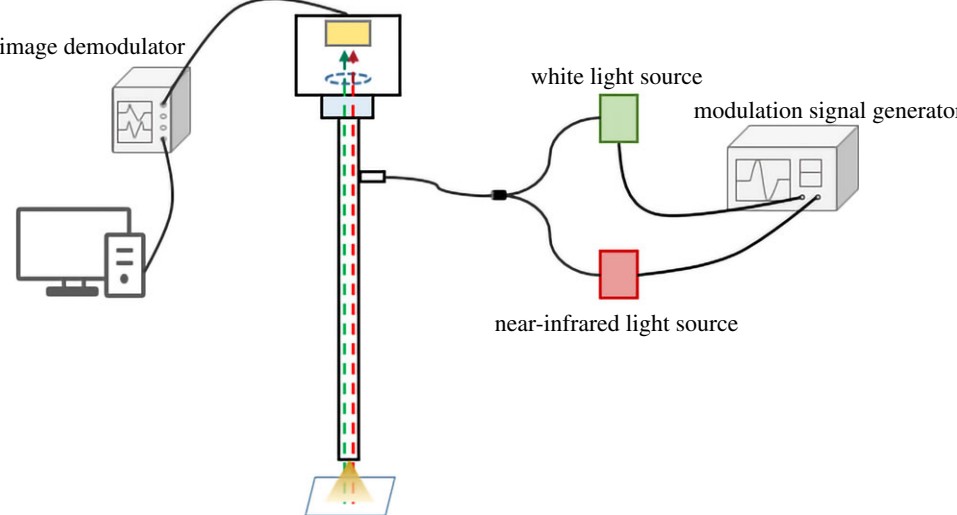

**Figure 3.** Schematic diagram of the fluorescence endoscope imaging system.

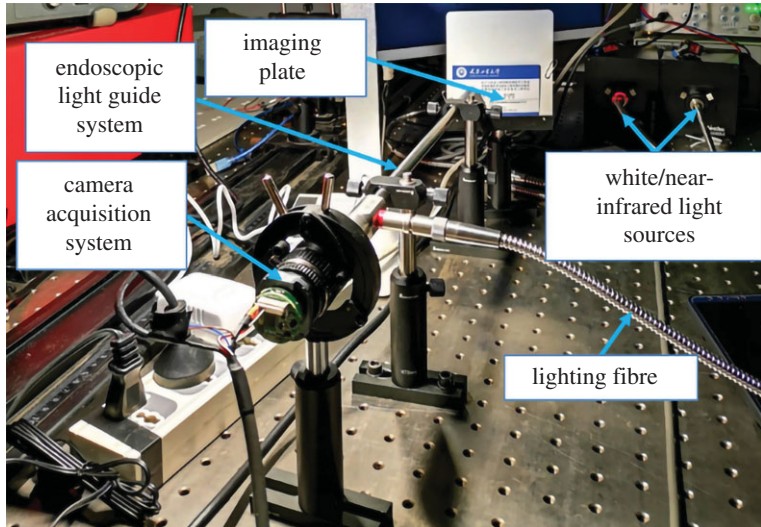

**Figure 4.** Fluorescence endoscopy imaging system.

without lock-in algorithm processing. Next, the image SNR is quantitatively calculated and analysed, and the results are shown in figure 6.

It can be seen from figure 6a and b that the SNR of the demodulated white and fluorescence images are relatively stable at the same noise level. Tables 1 and 2 show the SNRs of the white and the fluorescence images as functions of the noise level. As shown in tables 1 and 2, compared with the SNRs of white and fluorescence images without the lock-in algorithm processing, the SNRs of white and fluorescence images with the lock-in algorithm processing are effectively improved by about 7 dB.

## 4.2. Experiment results and analysis

In order to verify the availability of the digital lock-in algorithm in the fluorescence endoscopy imaging system, a fluorescence endoscopy imaging system was built. When using the fast digital lock-in algorithm, the sampling rate must be $4k$ times the original signal frequency, and $k$ is an integer ($f_s = 4k \cdot f$). In the experiment, the modulation frequency of the white light source was 12 Hz, the modulation frequency of the near-infrared light source was 6 Hz, and the acquisition frame rate of the camera was 48 fps. Without light source modulation and digital lock-in algorithm demodulation of the image, the time-sharing method is used to collect the white image and the fluorescence image respectively. Among them, the camera's acquisition frame rate is set to 48 fps, that is, within 1 s, 24

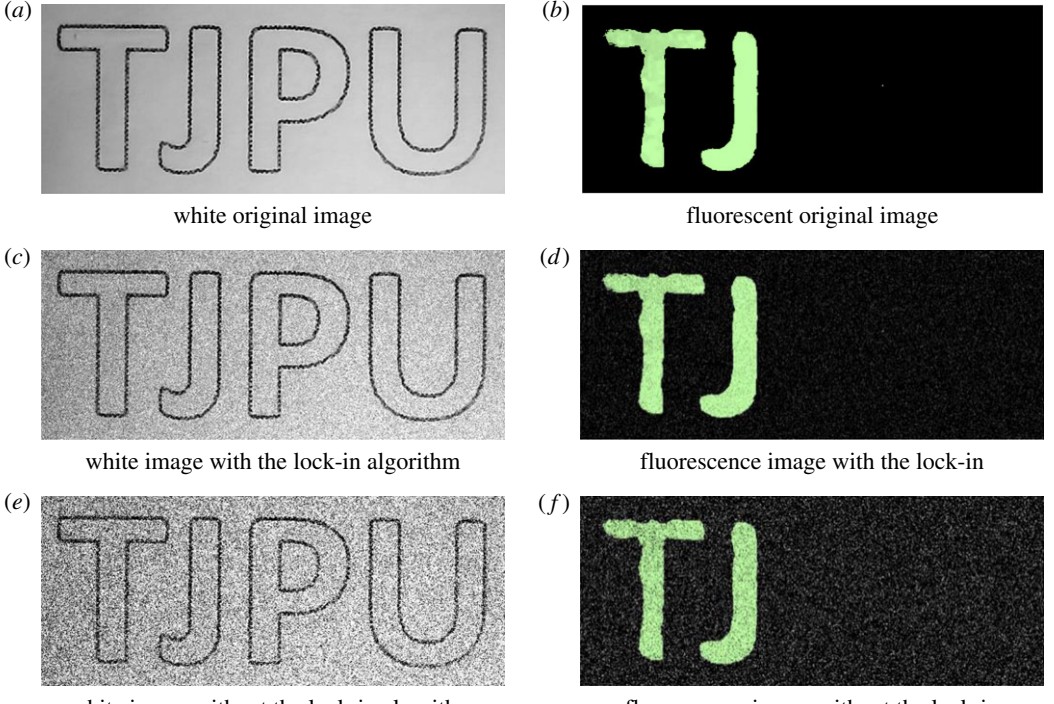

(a) white original image
(b) fluorescent original image
(c) white image with the lock-in algorithm
(d) fluorescence image with the lock-in
(e) white image without the lock-in algorithm
(f) fluorescence image without the lock-in

**Figure 5.** Original images, the images with the lock-in algorithm and the images without the lock-in algorithm. (a) White original image. (b) Fluorescent original image. (c) White image with the lock-in algorithm. (d) Fluorescence image with the lock-in algorithm. (e) White image without the lock-in algorithm. (f) Fluorescence image without the lock-in algorithm.

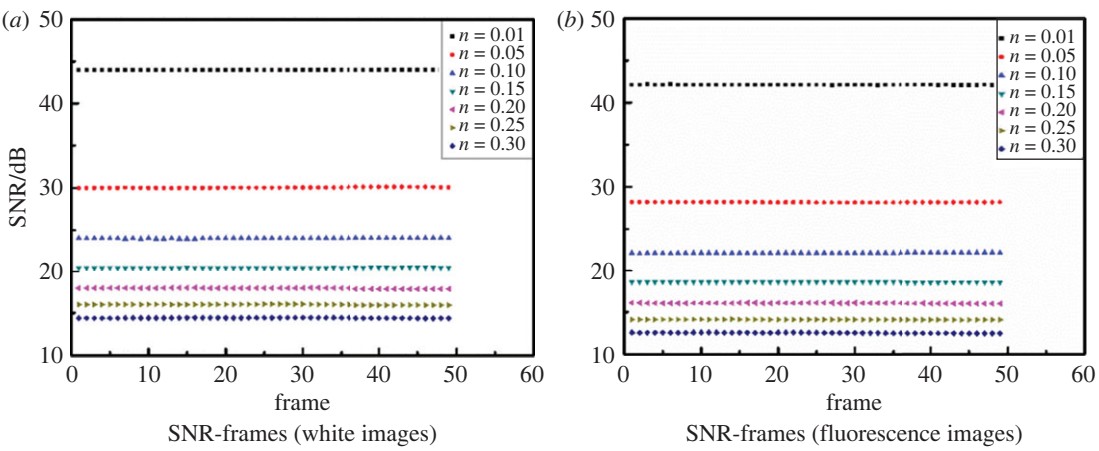

SNR-frames (white images)
SNR-frames (fluorescence images)

**Figure 6.** Image signal-to-noise ratio (SNR) analysis. (a) SNR-frames (white images). (b) SNR-frames (fluorescence images).

**Table 1.** White image signal-to-noise ratio (SNR) analysis.

| noise level | 0.01 | 0.05 | 0.1 | 0.15 | 0.2 | 0.25 | 0.3 |
|---|---|---|---|---|---|---|---|
| after lock-in image SNR/dB | 43.98 | 30.01 | 23.98 | 20.46 | 17.96 | 16.02 | 14.44 |
| without lock-in image SNR/dB | 36.98 | 23.01 | 16.99 | 13.46 | 10.97 | 9.03 | 7.45 |
| SNR improvement/dB | 7.00 | 7.00 | 6.99 | 7.00 | 6.99 | 6.99 | 6.99 |

frames of white images and 24 frames of fluorescent images are collected. In the experiment, a total of 88 white images and 88 fluorescent images were collected, and one white image and fluorescent image were extracted. Figure 7a and c show the white and the fluorescent images without the digital lock-in processing, respectively. Figure 7b and d show the white and the fluorescent images after the digital

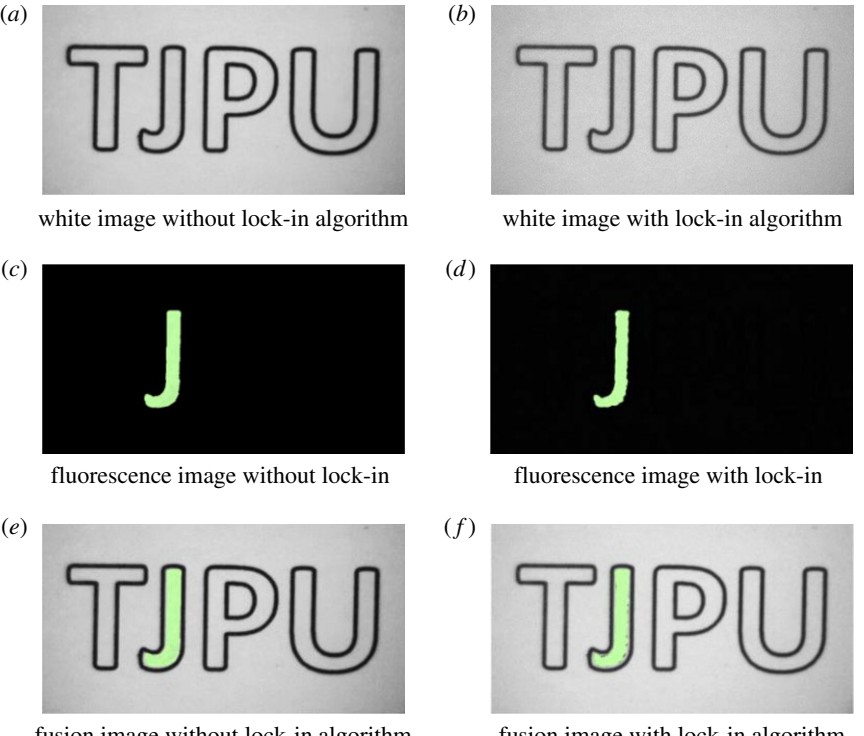

(a) white image without lock-in algorithm

(b) white image with lock-in algorithm

(c) fluorescence image without lock-in

(d) fluorescence image with lock-in

(e) fusion image without lock-in algorithm

(f) fusion image with lock-in algorithm

**Figure 7.** Images with and without lock-in algorithm. (*a*) White image without lock-in algorithm (*b*) White image with lock-in algorithm. (*c*) Fluorescence image without lock-in algorithm. (*d*) Fluorescence image with lock-in algorithm. (*e*) Fusion image without lock-in algorithm. (*f*) Fusion image with lock-in algorithm.

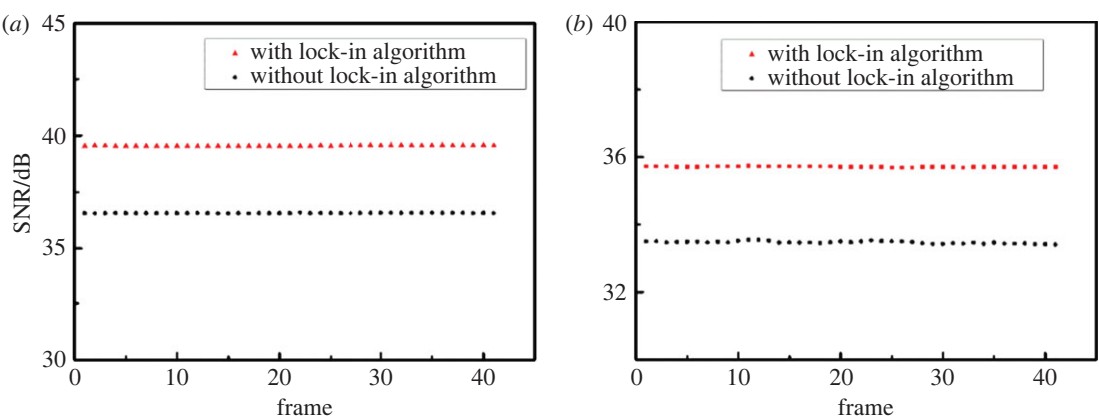

**Figure 8.** Image signal-to-noise ratio (SNR) analysis. (*a*) SNR-frame (white images). (*b*) SNR-frame (fluorescence).

**Table 2.** Fluorescence image signal-to-noise ratio (SNR) analysis.

| noise level | 0.01 | 0.05 | 0.1 | 0.15 | 0.2 | 0.25 | 0.3 |
|---|---|---|---|---|---|---|---|
| after lock-in image SNR/dB | 42.11 | 28.14 | 22.10 | 18.57 | 16.08 | 14.12 | 12.56 |
| without lock-in image SNR/dB | 35.11 | 21.13 | 15.11 | 11.59 | 9.09 | 7.15 | 5.57 |
| SNR improvement/dB | 7.00 | 7.01 | 6.99 | 6.98 | 6.99 | 6.97 | 6.99 |

lock-in processing, respectively. In figure 7*e* and *f*, fusion images of white and fluorescence images with and without the digital lock-in processing, are shown, respectively.

After that, the SNR of the collected white and fluorescence images were quantitatively calculated, and the results are shown in figure 8.

Figure 8*a* and *b* show the SNRs of the white and fluorescence images with and without digital lock-in algorithm processing, respectively. It can be seen from figure 8 that the SNRs of the images processed by the digital lock-in algorithm are stable. The average SNRs of white and fluorescence images without the digital lock-in algorithm processing are 36.56 dB and 33.47 dB, respectively, while the average SNRs of white and fluorescence images with the digital lock-in algorithm processing are 39.54 dB and 35.70 dB, respectively. Compared with the SNRs of white and fluorescence images without the lock-in algorithm processing, the SNRs of white and fluorescence images with the lock-in algorithm processing increased by 8.2% and 6.7%, respectively. Therefore, the fast digital lock-in algorithm can effectively enhance the image SNR and improve image quality.

# 5. Conclusion

In this paper, the white light source and the near-infrared light source of the fluorescence endoscope were modulated simultaneously, and fast image acquisition was performed. Then, the captured images were demodulated into white and fluorescent images using the fast digital lock-in algorithm, and the image SNR was quantitatively analysed. The experimental results showed that the fast lock-in algorithm can effectively improve the imaging quality of white and fluorescent images without changing the video output frequency. Compared with the current two-channel fluorescence endoscope obtaining white and fluorescence images based on a time-sharing method, the frequency division method has certain advantages in both the video output frequency and the SNR. Moreover, with the development of high-speed sampling optical devices, the imaging quality of the fluorescence endoscope based on the digital lock-in algorithm will be further improved and the development of a multi-spectral endoscope will be realized in the future. On the other hand, the fluorescence endoscope based on a fast digital lock-in algorithm can help further image processing, such as tumour edge detection, improve the accuracy of input data in big data analysis and ultimately can help or guide doctors to identify tumour regions in clinical practice.

Data accessibility. See the electronic supplementary material.

Authors' contributions. M.H. and F.X. performed the experiment, analysed results and drafted the manuscript. M.G. and S.Z. contributed to the experiment development. H.W. and J.W. conceived the study, analysed the results and supervised the project. G.H. and Z.Z. coordinated the study. All authors contributed to the writing of the manuscript and gave final approval for publication.

Competing interests. We have no competing interests.

Funding. This research was supported by the National Natural Science Foundation of China under grant no. 81901789 and the Tianjin Postgraduate Research and Innovation Project (2019YJSS013).

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
