## [Peer Review File · Royal Society Open Science]

Review History

RSOS-200779.R0 (Original submission)

Review form: Reviewer 1

Is the manuscript scientifically sound in its present form?

Yes

Are the interpretations and conclusions justified by the results?

No

Is the language acceptable?

Yes

Do you have any ethical concerns with this paper?

No

Have you any concerns about statistical analyses in this paper?

No

Recommendation?

Major revision is needed (please make suggestions in comments)

Comments to the Author(s)

The manuscript (RSOS-200779) improved the quality of white and fluorescence images obtained by fluorescence endoscope based on the digital lock-in algorithm. The proposed method was tested by the simulation experiment and the real system. The results show its performance. However, there are still many problems to be solved, and the manuscript still needs to be clarified in some places.

1. The authors can further emphasize the significance of the proposed method in the introduction, and compare their method and the results with related works in the last three years.
2. The modulation frequencies of the light sources are different. Can the authors describe how to choose the modulation frequencies?
3. For the calculation formula of the image SNR, what is the original image in the real system?
4. The authors should also provide more details about the simulation experiment.
5. Table 1 show the same information of Fig. 6(c) and 6(d). It can be removed.
6. The fusion image without lock-in algorithm in Fig. 7(e) looks like more sharper than the image in Fig. 7(f). Do you agree with me?
7. In the lock-in algorithm, how many images are used to get an improved image?
8. In the part of "Experiment results and analysis", there is little analysis, and the author mainly show the results.
9. In the title, the authors used fast digital lock-in algorithm, however, little information was provided in the manuscript.

Review form: Reviewer 2

Is the manuscript scientifically sound in its present form?

No

Are the interpretations and conclusions justified by the results?

Yes

Is the language acceptable?

No

Do you have any ethical concerns with this paper?

No

Have you any concerns about statistical analyses in this paper?

No

Recommendation?

Major revision is needed (please make suggestions in comments)

Comments to the Author(s)

No further comments to Author

Decision letter (RSOS-200779.R0)

Dear Mr Hu

The Editors assigned to your paper RSOS-200779 "Enhancement of Signal-to-noise Ratio for Fluorescence Endoscope Image Based on Fast Digital Lock-in Algorithm" have now received comments from reviewers and would like you to revise the paper in accordance with the reviewer comments and any comments from the Editors. Please note this decision does not guarantee eventual acceptance.

Please submit your revised manuscript and required files (see below) no later than 21 days from today's (ie 22-Oct-2020) date. Note: the ScholarOne system will 'lock' if submission of the revision is attempted 21 or more days after the deadline. If you do not think you will be able to meet this deadline please contact the editorial office immediately.

on behalf of the Associate Editor and Professor R. Kerry Rowe (Subject Editor)
openscience@royalsociety.org

Associate Editor Comments to Author:

With thanks to the referees for their commentary, we're of the view that a revision is required here. Each reviewer identifies a number of - possibly resolvable - matters including, for instance, the depth of analysis in the discussion and how your work is situated among more recent

research. There are also concerns regarding the parameterisation of algorithm. Please pay close attention to the reviewer comments in the decision letter and ensure that you not only fully respond to them but also include the changes requested in your revised paper. Furthermore, there are concerns regarding the quality of the written English: we sympathise as it is not a logical language, but services (<https://royalsociety.org/journals/authors/benefits/language-editing/>) exist to provide support - it is expected that you will seek language guidance and provide evidence of having done so. When you supply your revision, the reviewers will be invited to look again at the paper.

Reviewer comments to Author:

Reviewer: 1

Comments to the Author(s)

The manuscript (RSOS-200779) improved the quality of white and fluorescence images obtained by fluorescence endoscope based on the digital lock-in algorithm. The proposed method was tested by the simulation experiment and the real system. The results show its performance. However, there are still many problems to be solved, and the manuscript still needs to be clarified in some places.

1. The authors can further emphasize the significance of the proposed method in the introduction, and compare their method and the results with related works in the last three years.
2. The modulation frequencies of the light sources are different. Can the authors describe how to choose the modulation frequencies?
3. For the calculation formula of the image SNR, what is the original image in the real system?
4. The authors should also provide more details about the simulation experiment.
5. Table 1 show the same information of Fig. 6(c) and 6(d). It can be removed.
6. The fusion image without lock-in algorithm in Fig. 7(e) looks like more sharper than the image in Fig. 7(f). Do you agree with me?
7. In the lock-in algorithm, how many images are used to get an improved image?
8. In the part of "Experiment results and analysis", there is little analysis, and the author mainly show the results.
9. In the title, the authors used fast digital lock-in algorithm, however, little information was provided in the manuscript.

Reviewer: 2

Comments to the Author(s)

No further comments to Author

===PREPARING YOUR MANUSCRIPT===

Your revised paper should include the changes requested by the referees and Editors of your manuscript. You should provide two versions of this manuscript and both versions must be provided in an editable format:
 one version identifying all the changes that have been made (for instance, in coloured highlight, in bold text, or tracked changes);
 a 'clean' version of the new manuscript that incorporates the changes made, but does not highlight them. This version will be used for typesetting if your manuscript is accepted.
 Please ensure that any equations included in the paper are editable text and not embedded images.

===PREPARING YOUR REVISION IN SCHOLARONE===

Author's Response to Decision Letter for (RSOS-200779.R0)

See Appendix A.

RSOS-200779.R1 (Revision)

Review form: Reviewer 1

Is the manuscript scientifically sound in its present form?

Yes

Are the interpretations and conclusions justified by the results?

Yes

Is the language acceptable?

Yes

Do you have any ethical concerns with this paper?

No

Have you any concerns about statistical analyses in this paper?

No

Recommendation?

Accept as is

Comments to the Author(s)

Authors have revised the paper according to the reviewer's suggestion. Thus, this paper can be accepted.

Review form: Reviewer 2**Is the manuscript scientifically sound in its present form?**

Yes

Are the interpretations and conclusions justified by the results?

Yes

Is the language acceptable?

Yes

Do you have any ethical concerns with this paper?

No

Have you any concerns about statistical analyses in this paper?

No

Recommendation?

Accept as is

Comments to the Author(s)

Thank you for considering the review and alter the manuscript.

Decision letter (RSOS-200779.R1)

Dear Mr Hu,

It is a pleasure to accept your manuscript entitled "Enhancement of Signal-to-noise Ratio for Fluorescence Endoscope Image Based on Fast Digital Lock-in Algorithm" in its current form for publication in Royal Society Open Science. The comments of the reviewer(s) who reviewed your manuscript are included at the foot of this letter.

You can expect to receive a proof of your article in the near future. Please contact the editorial office (openscience@royalsociety.org) and the production office

(openscience_proofs@royalsociety.org) to let us know if you are likely to be away from e-mail contact – if you are going to be away, please nominate a co-author (if available) to manage the proofing process, and ensure they are copied into your email to the journal.

on behalf of R. Kerry Rowe (Subject Editor)
openscience@royalsociety.org

Reviewer comments to Author:
Reviewer: 1

Comments to the Author(s)
Authors have revised the paper according to the reviewer's suggestion. Thus, this paper can be accepted.

Reviewer: 2

Comments to the Author(s)
Thank you for considering the review and alter the manuscript.

Appendix A

Manuscript ID: RSOS-200779 Type: research article

Title: Enhancement of Signal-to-noise Ratio for Fluorescence Endoscope Image Based on Fast Digital Lock-in Algorithm

Author: Huiquan Wang; Meng Hu;
Jinhai Wang; Tiangong University

Reviewer: 1

Comments to the Author(s)

The manuscript (RSOS-200779) improved the quality of white and fluorescence images obtained by fluorescence endoscope based on the digital lock-in algorithm. The proposed method was tested by the simulation experiment and the real system. The results show its performance. However, there are still many problems to be solved, and the manuscript still needs to be clarified in some places.

1. The authors can further emphasize the significance of the proposed method in the introduction, and compare their method and the results with related works in the last three years.

Author response: Thank you for your valuable comments. I have modified it in the manuscript. The following content has been added in the introduction:

The team of professor Gang Li from Tianjin University [11-12] proposed a fast digital lock-in algorithm based on the classic digital lock-in algorithm. This algorithm no longer needs multiplication in the positive cross-correlation operation, and the amplitude and phase of the signal can be calculated by the addition and subtraction of discrete time series. Utilizing this fast digital lock-in algorithm greatly reduces the amount of calculation and data storage, and reduces the algorithm's requirements on the microprocessor. At the same time, the microprocessor does not need to generate a reference signal with the same frequency as the input signal, which reduces the burden on the microprocessor and improves the speed of algorithm realization.

The following references have been added:

- [1] Mei Zhou, Gang Li, and Ling Lin. 2013. Fast digital lock-in amplifier for dynamic spectrum extraction. *Journal of Biomedical Optics*. 18.5,57003. (doi: 10.1117/1.JBO.18.5.057003)
- [2] Gang Li, Mei Zhou, Xiaoxia Li, Ling Li. 2013. Digital lock-in algorithm and parameter settings in multi-channel sensor signal detection. *Measurement Journal of the International Measurement Confederation*. 46.8,2519-2524. (doi: 10.1016/j.measurement.2013.05.014)

2. The modulation frequencies of the light sources are different. Can the authors describe how to choose the modulation frequencies?

Author response: Thank you for your valuable comments. I have modified it in the manuscript. The following content has been added to the first paragraph of "Experiment results and analysis":

When using the fast digital lock-in algorithm, the sampling rate must be $4k$ times the original signal frequency, and k is an integer ($f_s=4k \cdot f$). In the experiment, the camera's acquisition frame

rate is 48 fps, the modulation frequency of the white light source is 12 Hz, and the modulation frequency of the near-infrared light source is 6 Hz. If high-speed cameras are used in future research, the modulation frequency can be increased.

3. For the calculation formula of the image SNR, what is the original image in the real system?

Author response: Thank you for your valuable comments. In the real system, Fig. 7 (a) (b) are the white original image and the fluorescent original image.

The original image is the expected value. The average value of multiple measurements is close to the original image.

4. The authors should also provide more details about the simulation experiment.

Author response: Thank you for your valuable comments. I have modified it in the manuscript.

The following content has been added to the first paragraph of "Simulation experiment results and analysis":

In order to objectively evaluate the effect of the digital lock-in algorithm on the image SNR of the fluorescence endoscopy imaging system, the white original image and the fluorescence original image were collected separately in this study, and the fluorescent agent ICG was applied to the two letters "TJ" in "TJPU" in the original image. MATLAB was used to conduct the simulation research of the digital lock-in algorithm. The original white original image and the fluorescent original image were copied to generate multiple frames of images, and noise was added to the image to simulate the noise in the actual image signal.

5. Table 1 show the same information of Fig. 6(c) and 6(d). It can be removed.

Author response: Thank you for your valuable comments. I have modified it in the manuscript.

Fig. 6(c) and 6(d) can be deleted, we have deleted Fig. 6(c) and 6(d) in the manuscript.

6. The fusion image without lock-in algorithm in Fig. 7(e) looks like more sharper than the image in Fig. 7(f). Do you agree with me?

Author response: Thank you for your valuable comments. It can be seen from Fig. 8 that the SNRs of the images processed by the digital lock-in algorithm are stable. The average SNRs of white and fluorescence images without digital the lock-in algorithm processing are 36.56 dB and 33.47 dB, respectively. While the average SNRs of white and fluorescence images with the digital lock-in algorithm processing are 39.54 dB and 35.70 dB, respectively. Compared with the SNRs of white and fluorescence images without the lock-in algorithm processing, the SNRs of white and fluorescence images with the lock-in algorithm processing increased by 8.2% and 6.7%, respectively. Therefore, the fast digital lock-in algorithm can effectively enhance the image SNR and improve the image quality.

In addition, when comparing the pictures in Fig. 7(e)(f), you should not only pay attention to the fluorescent image part, but also pay attention to the background color of the picture and its signal-to-noise ratio. For example, Figure 5(d)(f) can clearly see the noise of the background color

7. In the lock-in algorithm, how many images are used to get an improved image?

Author response: Thank you for your valuable comments. The demodulation process of lock-in technology is to down-sample 8 pictures into 1 picture. In the experiment, the camera's acquisition frame rate is set to 48 fps, that is, within 1 s, 48 frames of white light images and 48 frames of fluorescence images can be obtained. In this article, 88 frames of composite images are collected. For the white light image channel, the modulation signal frequency is 12 Hz, the amplitude is 1.5~2 V, and 88 frames of image signals can be obtained. For the fluorescence image channel, the modulation signal frequency is 6 Hz and the amplitude is 1.5~2 V, and 88 frames of image signals can be obtained.

8. In the part of "Experiment results and analysis", there is little analysis, and the author mainly show the results.

Author response: Thank you for your valuable comments. I have modified it in the manuscript. The following content has been added to the first paragraph of "Experiment results and analysis": Without light source modulation and digital lock-in algorithm demodulation of the image, the time-sharing method is used to collect the white image and the fluorescence image respectively. Among them, the camera's acquisition frame rate is set to 48 fps, that is, within 1 s, 24 frames of white images and 24 frames of fluorescent images are collected. In the experiment, a total of 88 white images and 88 fluorescent images were collected, and one white image and fluorescent image were extracted.

9. In the title, the authors used fast digital lock-in algorithm, however, little information was provided in the manuscript.

Author response: Thank you for your valuable comments. I have modified it in the manuscript. We have added a description of the fast lock-in algorithm in the introduction, and added relevant references.